# A series of magnon crystals appearing under ultrahigh magnetic fields in a kagomé antiferromagnet

R. Okuma [1], D. Nakamura[1], T. Okubo[2], A. Miyake[1], A. Matsuo[1], K. Kindo[1], M. Tokunaga [1], N. Kawashima[1], S. Takeyama[1] & Z. Hiroi[1]

Geometrical frustration and a high magnetic field are two key factors for realizing unconventional quantum states in magnetic materials. Specifically, conventional magnetic order can potentially be destroyed by competing interactions and may be replaced by an exotic state that is characterized in terms of quasiparticles called magnons, the density and chemical potential of which are controlled by the magnetic field. Here we show that a synthetic copper mineral, Cd-kapellasite, which comprises a kagomé lattice consisting of corner-sharing triangles of spin-1/2 $Cu^{2+}$ ions, exhibits an unprecedented series of fractional magnetization plateaus in ultrahigh magnetic fields of up to 160 T. We propose that these quantum states can be interpreted as crystallizations of emergent magnons localized on the hexagon of the kagomé lattice.

[1] Institute for Solid State Physics, The University of Tokyo, Kashiwa, Chiba 277-8581, Japan. [2] Department of Physics, The University of Tokyo, Tokyo 113-0033, Japan. Correspondence and requests for materials should be addressed to R.O. (email: rokuma@issp.u-tokyo.ac.jp)

Quantum many-body systems accommodate various exotic states and phenomena. One of the most notable examples is Bose–Einstein condensation (BEC), where a macroscopic number of bosonic particles occupy a single particle state as in a superfluid state of liquid $^4$He and in cold atomic gases. With the aid of attractive force, fermions in pairs can also condensate as in a superconducting state of electrons. In most antiferromagnetic insulators, the elementary excitation is a bosonic excitation magnon, and this can form a BEC[1–3]. Interestingly, interactions between magnons and couplings with the basal crystalline lattice lead to rich physics in quantum antiferromagnets, thereby distinguishing it from the canonical BEC.

The magnon picture has proven extremely fruitful for several antiferromagnets composed of spin-1/2 pairs with a spin-singlet ($S = 0$) ground-state, and triplet ($S = 1$) excitations called triplons. The triplons are similar to conventional magnons excited in an ordered antiferromagnet because both carry the spin angular momentum of $\hbar$, and thus the two terms are occasionally used interchangeably[2,3]. At a critical applied magnetic field, the energy of one of the Zeeman-split triplet components intersects the ground-state singlet, thereby resulting in a long-range magnetic order. Specifically, the transition corresponds to a BEC of diluted triplons (magnons), and this is typically observed in TlCuCl$_3$[4,5]. Above the critical field, the magnetization starts to increase linearly when the density of magnons increases with magnetic field. The magnetic field acts as a chemical potential for magnons, and thus controls the density of the magnons (which is proportional to the magnetization).

In simple spin systems, the magnetization increases smoothly with the magnetic field and eventually saturates. However, in certain quantum magnets, flat regions termed as magnetization plateaus appear at fractional magnetizations before saturation. There are two types of magnetization plateaus: a classical one that is described by a collinear arrangement of classical spins and a quantum state comprising entangled spins[6]. Classical magnetization plateaus are observed in triangular magnets, such as Cs$_2$CuBr$_4$[7] and Ba$_3$CoSb$_2$O$_9$[8,9], and the quantum plateaus in dimer magnets such as NH$_4$CuCl$_3$[10] and SrCu$_2$(BO$_3$)$_2$[11].

A transition to a quantum plateau as a function of magnetic field is considered to be a superfluid-insulator transition of hardcore bosons (magnons). Interacting magnons in a BEC state tend to localize due to the suppression of kinetic energy and eventually crystallize to become "insulating" such as Mott insulators in strongly correlated electron systems[12]. The magnon crystal exhibits a fixed density of magnons, and thus the magnetization remains at a fractional value of the full magnetization in a field range[6]. The fractional value of magnetization is attributed to the commensurability of the magnon crystal when there is no topological order. The number of magnons, $Q_{mag}S(1 - m)$, in the magnetic unit cell should be an integer where $Q_{mag}$, $S$, $m$ denote the number of spins in the magnetic unit cell, the spin quantum number, and the magnetization divided by the saturation magnetization, respectively[13]. In SrCu$_2$(BO$_3$)$_2$, which comprises pairs of Cu$^{2+}$ ions arranged orthogonally to each other in the sheet to form a Shastry–Sutherland lattice[11], a series of magnetization plateaus appear at $m = 1/8, 1/4, 1/3$ ($Q_{mag} = 16, 8, 12$)[14,15]; and nuclear magnetic resonance measurements directly confirmed spontaneous translational symmetry breaking in the magnon crystals[16].

In the spin-1/2 kagomé antiferromagnet (KAFM)[17–19], the ground-state is a gapless or gapful spin liquid and the formation of nontrivial magnons is theoretically expected immediately below the saturation[20]. When the magnetic field is set to infinitesimally smaller than the saturation field $B_s$, a magnon with total $S^z = 2$ in a hexagonal plaquette is generated in the fully polarized

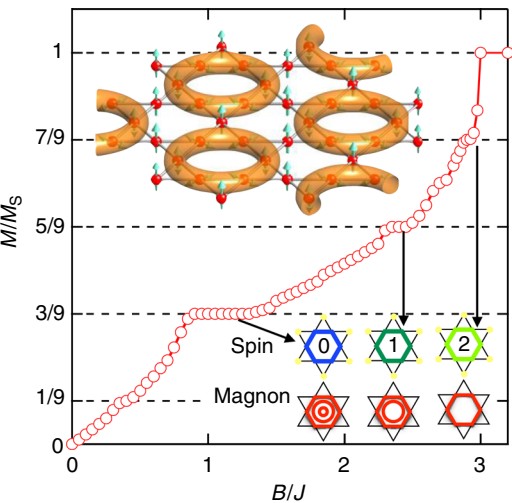

**Fig. 1** Calculated magnetization process for the spin-1/2 KAFM with the nearest-neighbor interaction $J$. The tensor network method with the projected entangled pair state (PEPS) is used. The vertical and horizontal axes represent magnetization $M$ divided by saturated magnetization $M_s$ and magnetic field $B$ divided by $J$, respectively. The top left inset shows a schematic drawing of hexagonal magnons that are depicted by doughnuts containing six entangled spins. The other intervening spins point upward in the direction of magnetic field. The magnon crystal forms a superlattice with a $\sqrt{3} \times \sqrt{3}$ unit cell. The bottom right inset shows hexagonal magnons expected to appear at the 1/3, 5/9, and 7/9 plateaus. In the upper part, the magnons are defined by the total spin $S^z = 0$, 1, and 2 for the six spins on the hexagon, respectively, while in the lower part based on the magnon picture, the number of magnons correspond to 3 (hexagon + double circle), 2 (single circle), and 1 (only hexagon), respectively. Using the bracket notation, the one-magnon state with $S^z = 2$ is expressed as $\sum_{i=1}^{6}(-1)^i S_i^-|0\rangle$, where the sum is obtained inside the hexagon, and $|0\rangle$ denotes the saturated state

spin state, which is the vacuum of magnons as schematically depicted in Fig. 1. Each spin inside the hexagonal plaquette equally carries fractional magnetization, and thus, the 'hexagonal magnon' corresponds to a highly quantum mechanical entity. Given the absence of energy cost for magnon generation, the density rapidly increases to 1/9 before the magnons overlap with each other to feel mutual repulsion. This results in an decrease in the magnetization from 1 to 7/9 at $B_s$[20]. Subsequently, a crystalline phase with a superstructure of the $\sqrt{3} \times \sqrt{3}$ unit cell with $Q_{mag} = 9$ is formed in a range of fields, thereby yielding a 7/9 magnetization plateau. A large magnon is emergently generated on a hexagon of the kagomé lattice in the KAFM, which is significantly different from dimer magnets with singlet and triplet states that naturally occur on built-in pairs of Cu ions.

Here, we report the observation of a series of fractional magnetization plateaus in the kagomé antiferromagnet Cd-kapellasite (CdK) and demonstrate the presence of emergent hexagonal magnons in the kagomé lattice. Some of the observed magnetization plateaus are reproduced by theoretical calculations for the simple KAFM model, while the others may be stabilized by lattice commensurability, additional long-range interactions, and potentially coupling to lattice.

## Results

**Theoretical predictions for multiple plateaus.** Recent calculations by the density-matrix-renormalization-group method, the exact diagonalization, and the tensor network method show that in addition to the well-established 7/9 plateau, three plateaus

appear at $m = 5/9$, 1/3, 1/9[21-24]. It is also suggested[21-23] that the 5/9 and 1/3 plateaus are magnon crystals with $Q_{mag} = 9$, which are similar to that at $m = 7/9$ but with $S^z = 1$ and 0, respectively, as depicted in Fig. 1; the 1/9 plateau is supposed to be another state, possibly a spin liquid with topological order[21]. However, one must be careful in concluding this because there are many competing phases in highly frustrated systems. In the present study, we calculated the complete magnetization process in the

extended projected entangled pair states (PEPS) scheme, and found three plateaus at $m = 1/3$, 5/9, 7/9, as shown in Fig. 1, which supports the previous results as shown in Fig. 1. Additionally, we examined the magnetic structure at the 1/3 plateau and observed that a magnon crystal composed of hexagonal magnons with $S^z = 0$ in the $\sqrt{3} \times \sqrt{3}$ structure was associated with the lowest energy among other competing states such as the up-up-down state (Supplementary Fig. 1). Thus, the hexagonal magnon must correspond to an entity that should be realized under the magnetic field for the KAFM, and this should be experimentally evidenced.

**Experimental obstacles.** Despite the intriguing predictions for the KAFM in magnetic fields, there is a paucity of experimental evidence due to the lack of ideal model compounds, and also the difficulties in experiments under high fields. Real materials always suffer from lattice distortion[25] or disorder[26], and this tends to mask the intrinsic magnetism of the KAFM. The best-characterized $S = 1/2$ KAFM is herbertsmithite with a large interaction of $J \sim 200$ K[27], and this implies that ultrahigh magnetic fields not $< B_s = 400$ T are necessary to record the complete magnetization process. However, experimentally available static magnetic fields are only below 45 T and typical pulsed magnetic fields are limited to below 100 T[28]. Furthermore, magnetization measurements by the conventional induction method are only available in pulsed magnetic fields below 100 T in the short time duration corresponding to several milliseconds;[28] for a larger pulsed field, a much shorter time window of several microseconds is available, which causes a large electromagnetic noise preventing

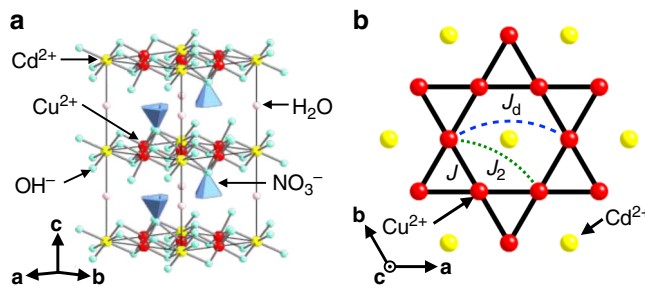

**Fig. 2** Crystal structure and magnetic interactions of CdK. **a** Kagomé layers comprising $Cu^{2+}$ and $Cd^{2+}$ ions possess good two dimensionality and are separated by nonmagnetic nitrate ions and crystalline water molecules. **b** Regular kagomé lattice composed of spin-1/2 $Cu^{2+}$ ions with a Cd ion in the center of each hexagon. Magnetic interactions between spins are nearest-neighbor $J$, a Dzyaloshinskii–Moriya (DM) interaction as large as 10% of $J$, small next-nearest-neighbor interaction $J_2$, and diagonal interaction $J_d$ across the Cd ion in the hexagon

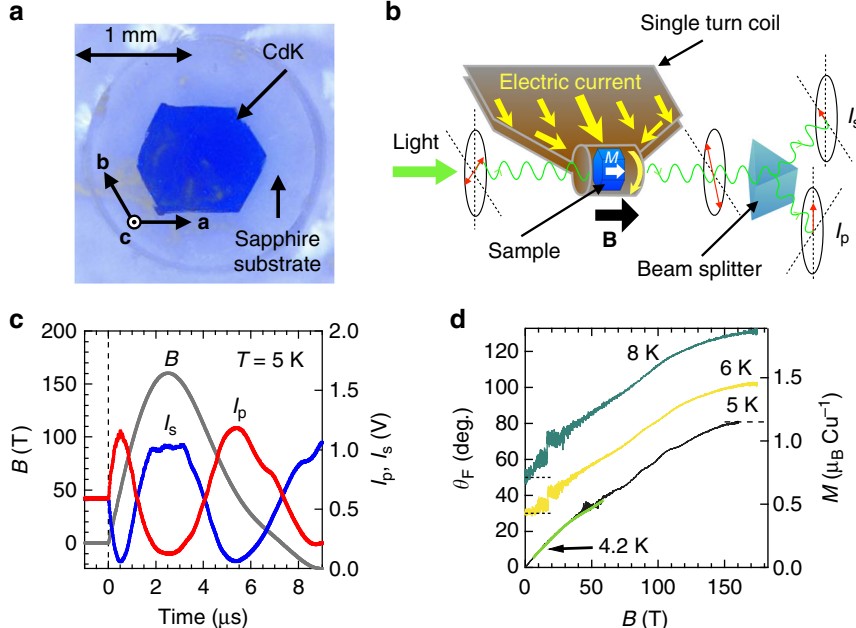

**Fig. 3** Magnetization measurements by the Faraday rotation technique in pulsed magnetic fields up to 160 T. **a** Single crystal of CdK on a sapphire disk substrate used in high-field Faraday rotation measurements. The crystal exhibits a flat (001) surface that effectively reduces scattering of light and is sufficiently large to obtain a Faraday rotation signal. **b** Schematic illustration of the experimental set-up. A high-magnetic field is generated inside a single-turn coil of 12 mm inner diameter by injecting a huge electric current (2–3 mega-ampere). **c** Typical time evolutions of a pulsed magnetic field with a maximum at 160 T and a time duration of 7 μs and $I_p$ and $I_s$ at 5 K. The hexagonal plate-like single crystal of CdK shown in **a** mounted on a sapphire substrate is placed in the coil such that the magnetic field is perpendicular to the hexagonal surface (**B** /// **c**). Linearly polarized incident light comes from the left and passes through the crystal. A change in the polarization angle due to the Faraday effect in the magnetized crystal is measured: transmitted light is split into vertical and horizontal components, $I_p$ and $I_s$, and each intensity is recorded by an oscilloscope. **d** Field dependences of the Faraday rotation angle $\theta_F$ and the corresponding magnetizations at 5 K (purple), 6 K (yellow), and 8 K (blue-green). For clarity, offsets of 20 and 50 degrees are added to the data at 6 and 8 K, respectively. The magnetization data from the Faraday rotation angle are calibrated to reproduce those obtained by the induction method by using a nondestructive pulse magnet as denoted by the black line below 55 T at 4.2 K. The large noises at approximately 30 T in the data at 6 and 8 K and at approximately 50 T in the data at 5 K are due to reductions in intensity for either $I_p$ or $I_s$ at the rotation angle corresponding to 90 degree

facile magnetization measurements. On the other hand, a state-of-the-art measurement using a Faraday rotation technique made it possible to record the complete magnetization process of $CdCr_2O_4$ up to 140 T, and this revealed a spin-nematic phase just below the saturation field[29]. In order to unveil the physics of the KAFM, both a suitable model compound with a relatively low-saturation field around 100 T and an improvement in the Faraday rotation technique are necessary. It is noted that the required low-temperature condition, $T \ll J/k_B$, is difficult to achieve for a compound with small $J$ and $B_s$ values.

**Kagomé antiferromagnet CdK.** To study the magnetism of $S = 1/2$ KAFM under magnetic fields, we selected Cd-kapellasite (CdK), $CdCu_3(OH)_6(NO_3)_2 \cdot H_2O$[30], which is isostructural to kapellasite, $ZnCu_3(OH)_6Cl_2$[31]. The compound has a quasi-two-dimensional structure with an undistorted kagomé lattice of $Cu^{2+}$ ions and a moderate antiferromagnetic interaction of $J \sim 45$ K ($B_s \sim 100$ T) (Fig. 2a)[32]. The ground-state of CdK is not a spin liquid but a long-range order (LRO) with a $\mathbf{q} = 0$ structure (a negative vector chirality order) below $T_N \sim 4$ K, which must be induced by a Dzyaloshinskii–Moriya (DM) interaction with a magnitude of approximately 10% of $J$. Other possible perturbations are the next-nearest-neighbor interaction $J_2$ and diagonal interaction $J_d$ bridged via the Cd ion in the center of the hexagon as shown in Fig. 2b, which was observed as a low value (Supplementary Fig. 2).

**Faraday rotation measurements up to 160 T.** We measured small magnetization from a tiny hexagonal single crystal of CdK with a diameter of approximately 1 mm and a thickness of 150 μm (Fig. 3a) by a Faraday rotation technique optimized to a single-turn coil in magnetic fields of up to 160 T. The

experimental set-up and results are shown in Fig. 3b. The polarization angle of an incident light with $\lambda = 532$ nm (Supplementary Fig. 3) was rotated with a varying magnetic field in several microseconds via the Faraday effect from induced magnetization in the sample (Fig. 3c). Figure 3d shows magnetizations converted from the data in descending pulsed magnetic fields (Supplementary Fig. 4). The magnetization curve measured at 8 K smoothly increased and saturated at approximately 150 T. At 6 K, a faint wiggling was observed, which seemed to produce many anomalies at 5 K.

Figure 4a shows the magnetization process at 5 K and its field derivative, which is compared with the calculated magnetization process in the presence of DM interaction. Below 0.4 $\mu_B$, the experimental magnetization is in good agreement with the calculation and deviates above that. At least seven anomalies are observed at magnetizations exceeding 0.4 $\mu_B$. Above $B_s = 160$ T, $M$ reaches 1.15 $\mu_B$/Cu, and this was nearly equal to the saturation magnetization with fully polarized spins: $gS\mu_B$ with $g_c \sim 2.3$ from magnetic susceptibility (Supplementary Fig. 2).

**Magnetization plateaus.** We consider the anomalies as a series of blunt plateaus that were blurred or inclined due to the finite temperature effect or anisotropy such as DM interaction. In conventional ordered antiferromagnets a metamagnetic transition or a spin-flop transition can occur in a magnetic field. However, weak anisotropy in $Cu^{2+}$ spin systems makes it difficult for such transitions to occur. We consider blunting by temperature and define a critical field $B_k$ and a magnetization $M_k$ for each plateau at which the differential magnetization takes a local minimum; the $B_k$ should correspond to the center of the field range for the plateau. The obtained values of $[B_k$ (T), $M_k$ ($\mu_B$/Cu), $m_k]$ ($m_k = M_k/1.15$) are (47.6, 0.42, 0.37), (72.6, 0.59, 0.51), (95.3,

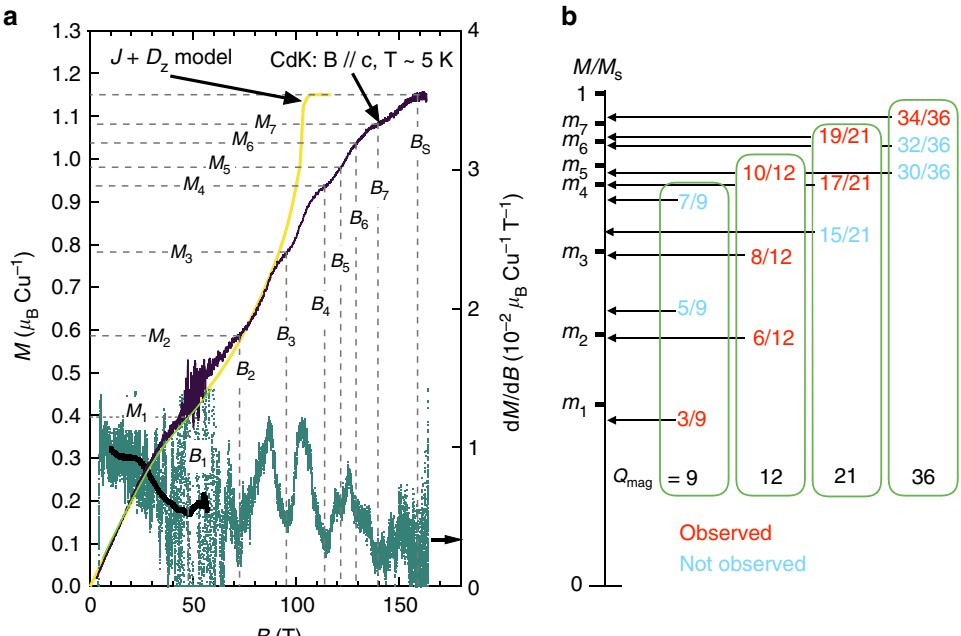

**Fig. 4** Appearance of multiple magnetization plateaus in CdK. **a** Field dependences of magnetization and its derivative at 5 K and **B // c**. The purple and yellow-green lines represent magnetizations measured by the Faraday rotation and induction method, respectively, and blue-green and black points denote the field derivative of the former and latter curve, respectively. The magnitudes of $B$ and $M$ at seven local minima in the field derivative curve are termed as $B_k$ and $M_k$ as shown by the broken lines, which can correspond to a series of magnetization plateaus. The saturation field $B_s = 160$ T is determined at which $M$ reaches the full magnetization $M_s = 1.15$ $\mu_B$. The green line shows the $M$ calculated by PEPS for an $S = 1/2$ KAFM model with $J = 45$ K and the $z$ component of DM interaction of 0.1$J$. **b** Comparison between experimentally observed plateau magnetizations $m_k = M_k/M_s$ and fractional magnetizations expected from the series of magnon crystals with $Q_{mag} = 9, 12, 21, 36$ unit cells as shown in Fig. 5. The fractions in the red bold letter are close to one of the $m_k$ values while the fractions in regular blue font cannot be observed with the exception of 30/36 that is assigned to 10/12

0.78, 0.68), (113.8, 0.94, 0.81), (121.6, 0.98, 0.85), (129.1, 1.04, 0.9), and (139.7, 1.08, 0.94) for $k = 1$–$7$, respectively. It should be noted that the anomalies are distinct for $k = 2$, 3, 4, and 7 and weaker for $k = 5$ and 6. As mentioned above, it is expected that magnetization plateaus from magnon crystals with $Q_{mag} = 9$ appear only at $m = 1/3$, 5/9, and 7/9 for the simple KAFM. The values were near to $m_1$, $m_2$, and $m_4$, as shown in Fig. 4b, although they significantly deviated from them. The present observation of the increased number of plateaus in CdK significantly indicate the formation of other types of magnon crystals with unit cells exceeding $Q_{mag} = 9$.

## Discussion

We considered possible magnetic structures realized at the magnetization plateaus in CdK. As in the case of the simple KAFM, we assumed that a type of localized hexagonal magnons is periodically aligned to reduce mutual repulsion while maintaining a sixfold rotational symmetry. Subsequently, such a series of magnon crystals as shown in Fig. 5 with unit cells of $Q_{mag} = 9$, 12, 21, 36 can appear wherein each involves three sequences with 1–3 magnons on the hexagon. All the experimentally observed $m_k$ values are well reproduced. For example, $m_7 = 0.94$ near the saturation is close to $34/36 \sim 0.944$, and $m_6 = 0.90$ is nearly equal to $19/21 \sim 0.905$. Furthermore, $m_5 = 0.85$, $m_4 = 0.82$, $m_3 = 0.68$, $m_2 = 0.51$, and $m_1 = 0.37$ are near $5/6 \sim 0.833$, $17/21 \sim 0.810$, $2/3 \sim 0.667$, $1/2 = 0.5$, and $1/3 \sim 0.333$, respectively (Fig. 4b). Thus, one from the $Q_{mag} = 9$ series, three from $Q_{mag} = 12$, two from $Q_{mag} = 21$, and one from $Q_{mag} = 36$ were observed. The fact that all the three sequences appeared for $Q_{mag} = 12$ while only part appeared for the others prompted us to determine a rule for the magnon crystallization in CdK.

Emergence of larger unit cells in CdK must be attributed to additional farther–neighbor interactions such as $J_2$ and $J_d$. Our calculations of magnetization without them and with 10% DM interaction does not exhibit any anomalies after an inclined 1/3 plateau with increasing fields (Fig. 4a). It is emphasized that the anomalies in CdK appear when the experimental magnetization curve deviates from the theoretical one, which suggests that effects excluded in the calculation play a crucial role. If $J_2$ works as an effective repulsion among magnons, the $Q_{mag} = 9$ structure must become unstable (Supplementary Fig. 5). For the $Q_{mag} = 12$ structure, on the other hand, $J_2$ does not work while $J_d$ does. The experimental fact that all the three sequences are observed for the $Q_{mag} = 12$ series and only a part for the others (e.g., not 5/9 but 6/12 for $m_2$) indicates the relative stability of $Q_{mag} = 12$. This implies repulsive (ferromagnetic) $J_2$ and attractive (antiferromagnetic) $J_d$. Larger unit cells, such as $Q_{mag} = 21$ and 36, can be stabilized by weak farther–neighbor interactions or the enlargement of the effective size of hexagonal magnons. Therefore, the appearance of multiple magnetization plateaus in CdK is reasonably explained by assuming a series of magnon crystals in a spin-1/2 KAFM with additional interactions.

Finally, we focus on the reason for the appearance of magnetization plateaus in CdK after the deviation from the calculation based on the $J +$ DM model; the smooth rise of the calculated curve indicates that spins are gradually forced to align along the magnetic field from the coplanar $\mathbf{q} = 0$ structure as expected for a classical spin system. One possibility is the magnetostriction that induces a lattice distortion[33]. For example, a hexagon where a localized magnon lives can deform to increase antiferromagnetic $J$, and thereby to stabilize a local magnon. When this type of a lattice distortion occurs periodically to form a superstructure, hexagonal magnons tend to crystallize at high fields with $m \geq 1/3$. Therefore, a spin-lattice coupling potentially makes a tilted $\mathbf{q} = 0$ magnetic structure unfavorable and leads to a series of

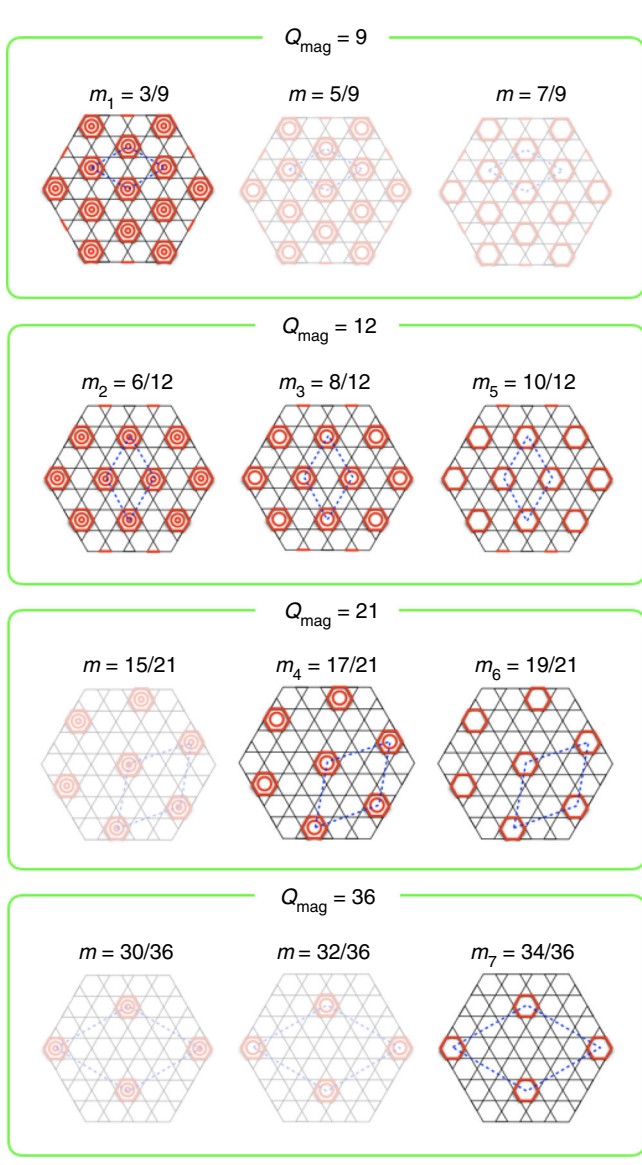

$S^z = 0$ $S^z = 1$ $S^z = 2$

**Fig. 5** Possible series of magnetic structures for the magnon crystals with $Q_{mag} = 9$, 12, 21, and 36. Among them, those corresponding to the observed seven magnetization plateaus are emphasized. Red hexagons denote localized hexagonal magnons with the total $S_z = 0$, 1, and 2. Other intervening spins are fully polarized along the magnetic field as shown for $Q_{mag} = 9$ in the inset of Fig. 1. Blue dotted lozenges represent magnetic unit cells

crystallizations of localized magnons with the aid of farther–neighbor interactions.

## Methods

**Sample preparation.** Single crystals of Cd-kapellasite, $CdCu_3(OH)_6(NO_3)_2 \cdot H_2O$, were synthesized by a two-step hydrothermal method[32]. A quartz ampoule with a length of 150 mm was filled with ingredients ($Cu(OH)_2$: 0.1 g, $Cd(NO_3)_2 \cdot 4H_2O$: 5 g, $H_2O$: 4 g) and sealed. A thick quartz tube with outer and inner diameters of 12 and 8 mm, respectively, was used to avoid bursting due to increased pressure inside the tube during heat treatment. The ampoule was placed horizontally in a two-zone furnace and heated to 180 °C at the hot end and to 130 °C at the cold end for a week. Polycrystalline samples of CdK were quickly produced over the tube and subsequently slowly transported into the cold zone to condense into a bunch of single crystals 100 μm in diameter. After all polycrystalline samples were transported to the cold zone, the temperature gradient was reversed. The ex-cool zone was set to 160 °C, and the ex-hot zone was set to 140 °C. This resulted in inverse

transportation, thereby resulting in the growth of hexagonal plate-like crystals as large as 1 mm in diameter and 150 μm in thickness as typically shown in Fig. 3a.

**Magnetization measurements**. Magnetization measurements in low-magnetic fields below 7 T were conducted by using a single crystal of CdK with a weight of 2.38 mg in a Magnetic-Property-Measurement-System 3 (MPMS-3, Quantum Design). Magnetic susceptibilities along the **a** and **c** axes were measured at the constant field of 1 T. Magnetization at pulsed magnetic fields up to 60 T was measured at 4.2 K on stacked single crystals via the electromagnetic induction method. The absolute value of magnetization was calibrated by the data obtained from MPMS-3.

**Faraday rotation measurements in pulsed magnetic fields**. A single-turn coil as shown in Fig. 3a was used to generate ultrahigh magnetic fields up to 160 T[34]. A sample was not destroyed after the experiment because the coil explodes outwards along the direction of Maxwell stress. Faraday rotation was measured in a short time duration corresponding to 7 μs. The sample temperature was set to 5, 6, and 8 K and could slightly change at elevated fields due to the magnetocaloric effect in the adiabatic condition. A change in the polarization angle $\theta_F$ was proportional to magnetization $M$ induced by the applied field in the sample: $\theta_F = \alpha M d$ where $\alpha$ denotes the Verdet constant and $d$ denotes the thickness of the sample. Specifically, $\theta_F$ is calculated from the intensities of the vertical and horizontal components, $I_p$ and $I_s$, by the formula: $\theta_F = \cos^{-1}\{(I_p - I_s)/(I_p + I_s)\}$.

**Tensor network calculations**. Magnetization process of a $S = 1/2$ KAFM with the nearest-neighbor interaction $J$ is calculated by the infinite projected entangled pair state (iPEPS)[35–37] method that expresses a wave function as an extended tensor product state termed as the projected simplex pair state (PESS)[38,39]. Unit cells up to 18 sites are assumed. The wave function is optimized by the simple update method[40,41] with bond dimensions $D = 4–7$. Physical quantities were typically calculated by the corner transfer matrix method[42] with bond dimensions $\chi = D^2$.

## Data availability
The publication data used in this study is available on https://doi.org/10.6084/m9.figshare.7609745.v1. The data files are also available from the corresponding author upon request. The source data underlying Figs. 1, 3c, d, and 4a, and Supplementary Figures 1–4 are provided as a Source Data file.

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

## Acknowledgements
We are grateful to T. Misawa, H. Tsunetsugu, and C. Hotta for helpful discussion. R.O. is supported by the Materials Education Program for the Future Leaders in Research, Industry, and Technology (MERIT) under the Ministry of Education, Culture, Sports, Science, and Technology of Japan (MEXT). The study was partially supported by KAKENHI (Grant No. 15K17701), the Core-to-Core Program for Advanced Research Networks under the Japan Society for the Promotion of Science (JSPS), and MEXT of Japan as a social and scientific priority issue (Creation of new functional devices and

high-performance materials to support next-generation industries; CDMSI) to be solved by using post-K computer and "Exploratory Challenge on Post-K computer" (Challenge of Basic Science–Exploring Extremes through Multi-Physics and Multi-Scale Simulations).

## Author contributions

R.O., D.N., S.T., and Z.H. conceived and designed the study. A. Miyake and A. Matsuo measured the magnetization curve up to 60 T under supervision of M.T.; K.K., R.O., D.N. measured the magnetization curve up to 180 T under supervision of S.T.; R.O., D.N., S.T., and Z.H. interpreted the experimental data. T.O. performed and interpreted PEPS calculations under supervision of N.K.; R.O. wrote the manuscript. All authors discussed and commented on the manuscript.

## Additional information

**Competing interests:** The authors declare no competing interests.

