## [Peer Review File · Nature Communications]

Reviewers' comments:

Reviewer #1 (Remarks to the Author):

This paper reports a new 'devil's staircase' compound showing a series of plateaus in the magnetization vs magnetic field. These measurements up to 160 T are performed on a kagome compound in which such a staircase has been calculated in this paper, and is based on a previous theory paper from 2002 referenced within. The devil's stair case consists of different ordered patterns of 6-spin states residing on hexagons. The results looks both valid and important to me. Devil's staircase compounds as well as Kagome systems are a current topic, and will generate interest and follow-on work. I recommend publishing in Nature Communication after considering the following minor comments:

1) There is some controversy over whether to use the term 'magnon' (Rev. Mod. Phys. 86, 563, 2014) since a magnon is an excitation, not a ground state of the system and magnons are generally propagating. The states being referenced in this paper are ground-state spin configurations of six entangled spins, not excitations. Researchers who study actual magnon states e.g. in pumped-probe systems (PRL 102, 187205 2009 and similar) have raised the issue that it creates confusion to call ground state spin configurations magnons in contrast to actual magnons. However I also appreciate that the referenced 2002 theory paper [18] that this work is based on does use the term 'magnon'. That paper predates the current terminology controversy. Thus the authors could add a note to this effect, namely that the 'magnons' here are not propagating excitations but rather ground-state spin configurations of the six-fold spin system on the kagome hexagon.

It would also be very helpful to a broad audience to define these "magnon" or "spin" states using bracket notation in terms of the states of the six individual spins. Otherwise it is difficult to follow what is meant by 'one' 'two' and 'three magnon' states.

Another option is 'minion' which is preferred by my spellchecker. (Just kidding.)

2) Given the fast microsecond speed of the pulses, one might wonder if the sample is at constant temperature or if there is magnetocaloric heating of the sample. This is difficult to ascertain in experiments, even though the up and down sweeps are similar. Temperature changes can be reversible. Such temperature changes could be estimated from theory. If there are significant temperature changes, they could account for some of the differences between theory and experiment. Alternatively if temperature changes might be significant, the data could be compared to an adiabatic calculation in which the temperature is allowed to change but but the sample is held at constant entropy.

3) Editing for grammar needs to be performed.

Reviewer #2 (Remarks to the Author):

The manuscript reported a series of field induced magnon crystals (or quantum spin state transitions) in a kagome aniferromagnet under ultra high magnetic fields. This is a rare example to experimentally observe such kind of magnetization plateaus related to the magnon crystals in a kagome system. This new result is expected to attract attentions in the quantum magnetism community and therefore the referee recommends its publication on Nature Communications.

Meanwhile, the referee feels the presentation of the manuscript could be improved before the formal publication, especially the introduction. Here are some comments:

(i) The referee feels that the introduction is not well organized and the messages are scattered around. The authors first talk about quantum spin liquid, then magnetic field, then BEC, then magnon in SrCuBO, then calculation of magnetization on kagome lattice and then references on ZnCr2O4 and GGG, and then the Farady rotation technique. Too many different things here, the referee encourages the authors to reorganize the paragraphs to make the messages that they want the readers to see more focused.

(ii) The authors cite several examples about magnetization plateau (again they cite them on different places in the introduction, but basically, they are same, all of them show magnetization plateaus), which are SrCuBO with square lattice, ZnCr2O4 with pyrochlore lattice and GGG with a 3D triangular lattice. While all these examples are good to show the studied magnetization plateaus, the authors missed another important lattice showing magnetization plateau, the 2D triangular lattice. In fact, the triangular lattice with edge shared triangles is the closest similar frustrated lattice to kagome (the corner-shared triangular lattice). Therefore, it makes more sense to cite the magnetization plateau in triangular antiferromagnets. For example, in the distorted triangular lattice Cs2CuBr4 (PRL 102, 257201 2009), Fortune et al, has already reported 1/3, 1/2, 5/9, 2/3 Ms plateaus, which is kind of similar to the results here on kagome lattice. Another good example is the Ba3CoSb2O9 (PRL 109, 267206 2012; PRL 108, 057205 2012) with equilateral triangular lattice showing 1/3Ms plateau.

Reviewer #3 (Remarks to the Author):

This manuscript reports the fractional magnetization plateaux experimentally observed in the kagome material Cd-kapellasite under ultrahigh magnetic fields. The highlight of this work is the observation of the whole magnetization process (thanks to the moderate nearest coupling $J \sim 45\text{K}$) and a series of magnetization plateau which cannot be explained by the standard Heisenberg model with Dzyaloshinskii-Moriya interactions. The authors further suggested the magnon crystal patterns which would explain the experimental observations and proposed a microscopic model with further-neighbor interactions (J_2 and J_d).

The experimental results reported in this manuscript are very interesting and should attract attentions from the condensed matter community (in particular, those researchers who work on frustrated magnets). However, the theoretical/numerical results in this manuscript are not fully satisfactory and, in my opinion, there is still room to improve. For this reason, I cannot recommend the manuscript for publication in its present form, unless the authors provide a convincing reply (and revise their manuscript, if needed).

My main concern is that the claim of necessary longer-range interactions (J_2 and J_d) are not fully justified by the results reported in the manuscript and, in my opinion, it is possible to achieve better results by using the numerical technique that the authors already have. Let me explain in more details below.

In Fig. 5, the authors proposed a possible series of magnon crystals for explaining the experimentally observed magnetization plateaux in Fig. 4. It is also suggested that ferromagnetic J_2 and antiferromagnetic J_d interactions would help to stabilize the magnon crystals in Fig. 5.

Since this is a main result, several issues need to be clarified:

1) Were there any ab initio calculations performed for this material? This would indicate how large J_2 and J_d would be.

2) Are there any indications from the crystal structure that J_2 and J_d are important? The lengths and angles of the bonds in this material have been reported in Ref. 29, which may give some hints.

3) Did the authors perform PEPS simulations based on the modified kagome model including J , J_2 , J_d , and D terms? I would be surprised if this has not been done, because the authors have calculated the magnetization curve for the Heisenberg model with DM interactions (see Fig. 1). Why cannot the authors calculate the magnetization curve for the modified kagome model including J , J_2 , J_d , and D terms? (By the way, the bond dimension $D=4-7$ in the PEPS simulations performed for obtaining Fig. 1 is pretty small compared to other PEPS simulations in the literature, e.g., those in Ref. 37. What is the reason for that?)

Below I also have two concrete comments on the manuscript:

1) In the last sentence of the abstract, the authors claimed that their observation "reveals a novel type of particle physics realized in a highly frustrated magnet". This is quite confusing. Why is it related to particle physics? I would say that magnons are just ordinary quasi-particles in solid state physics.

2) In the last paragraph of the main text, it is written that "we are now challenging a magnetization measurement at extremely high fields of 600T, We think that these results will offer not only new physics in frustrated magnetism but also new insights into ...". In my opinion, these claims are irrelevant for the present results (note that this is not a research proposal) and should be removed from the manuscript.

Response to Reviewer #1:

Thank you for a positive comment.

“The results looks both valid and important to me. Devil's staircase compounds as well as Kagome systems are a current topic, and will generate interest and follow-on work. I recommend publishing in Nature Communication after considering the following minor comments:”

Below we present our responses to his/her comments and questions shown by **R(n)** and **Q(n)**, respectively.

Q(1) There is some controversy over whether to use the term 'magnon' (Rev. Mod. Phys. 86, 563, 2014) since a magnon is an excitation, not a ground state of the system and magnons are generally propagating. The states being referenced in this paper are ground-state spin configurations of six entangled spins, not excitations. Researchers who study actual magnon states e.g. in pumped-probe systems (PRL 102, 187205 2009 and similar) have raised the issue that it creates confusion to call ground state spin configurations magnons in contrast to actual magnons. However I also appreciate that the referenced 2002 theory paper [18] that this work is based on does use the term 'magnon'. That paper predates the current terminology controversy. Thus the authors could add a note to this effect, namely that the 'magnons' here are not propagating excitations but rather ground-state spin configurations of the six-fold spin system on the kagome hexagon.

R(1) As the referee point out, the term “magnon” is traditionally used to indicate a spin wave excitation from a long-range-ordered magnetic state and now also used to imply a ground-state spin configuration. The hexagonal magnon discussed for the kagomé antiferromagnet in the present paper is the latter and can be the former only above the saturation field. To clarify this fact, we add the following sentence in the second paragraph of page 2 in the revised manuscript:

The magnon picture is proven as extremely fruitful in several antiferromagnets composed of pairs of spin-1/2 with a spin-singlet ($S = 0$) ground state and triplet ($S = 1$) excitations termed as triplons. The triplons are similar to conventional magnons excited in an ordered antiferromagnet because both carry the spin angular momentum of \hbar , and thus the two terms are occasionally used interchangeably^{2,3}.

Q(2) It would also be very helpful to a broad audience to define these "magnon" or "spin" states using bracket notation in terms of the states of the six individual spins. Otherwise it is difficult to follow what is meant by 'one' 'two' and 'three magnon' states.

R(2) We agree that the explicit form of the hexagonal magnons would help readers to understand them. However, such notation is available only in the case of “one” hexagonal magnon. Therefore, we add this sentence into the caption of Fig. 1:

Using the bracket notation, the one-magnon state with $S^z = 2$ is expressed as $\sum_{i=1}^6 (-1)^i S_i^- |0\rangle$, where the sum is obtained inside the hexagon and $|0\rangle$ denotes the saturated state.

Q(3) Given the fast microsecond speed of the pulses, one might wonder if the sample is at constant temperature or if there is magnetocaloric heating of the sample. This is difficult to ascertain in experiments, even though the up and down sweeps are similar. Temperature changes can be reversible. Such temperature changes could be estimated from theory. If there are significant temperature changes, they could account for some of the differences between theory and experiment. Alternatively if temperature changes might be significant, the data could be compared to an adiabatic calculation in which the temperature is allowed to change but the sample is held at constant entropy.

R(3) Our measurements were done in a nearly adiabatic condition. Thus, the sample temperature was increased at elevated fields. We have carried out magneto-caloric effect measurements up to 60 T for CdK, which reveals that the rise of temperature is smaller than 2 K, starting from 5 or 8 K at zero field. Since our experimental setup in the Faraday rotation measurements is so complex, it is not possible to estimate the temperature rise theoretically. However, our data have been already smeared out by thermal effects, which becomes more serious with increasing the starting temperature, as shown in Fig. 3, the temperature rise does not affect our observations. To clarify the meaning of the temperature in the experiments, the following note is added to the experimental section:

The sample temperature was set to 5, 6, and 8 K and could slightly change at elevated fields due to the magnetocaloric effect in the adiabatic condition.

Response to Reviewer #2:

Thank you for this positive comment.

“This is a rare example to experimentally observe such kind of magnetization plateaus related to the magnon crystals in a kagome system. This new result is expected to attract attentions in the quantum magnetism community and therefore the referee recommends its publication on Nature Communications.”

Below we reply to his/her comments and questions.

Q(1) The referee feels that the introduction is not well organized and the messages are scattered around. The authors first talk about quantum spin liquid, then magnetic field, then BEC, then magnon in SrCuBO, then calculation of magnetization on kagome lattice and then references on ZnCr₂O₄ and GGG, and then the Farady rotation technique. Too many different things here, the referee encourages the authors to reorganize the paragraphs to make the messages that they want the readers to see more focused.

R(1) Our central message is the localization and delocalization of emergent bosonic particle “hexagonal magnon.” As the referee suggested, the introduction contained too many general examples of quantum states such as spin liquid or GGG. We revised the introduction to focus on the bosonic picture of quantum magnets under magnetic fields in the following.

Q(2) The authors cite several examples about magnetization plateau (again they cite them on different places in the introduction, but basically, they are same, all of them show magnetization plateaus), which are SrCuBO with square lattice, ZnCr₂O₄ with pyrochlore lattice and GGG with a 3D triangular lattice. While all these examples are good to show the studied magnetization plateaus, the authors missed another important lattice showing magnetization plateau, the 2D triangular lattice. In fact, the triangular lattice with edge shared triangles is the closest similar frustrated lattice to kagome (the corner-shared triangular lattice). Therefore, it makes more sense to cite the magnetization plateau in triangular antiferromagnets. For example, in the distorted triangular lattice Cs₂CuBr₄ (PRL 102, 257201 2009), Fortune et al, has already reported 1/3, 1/2, 5/9, 2/3 Ms plateaus, which is kind of similar to the results here on kagome lattice. Another good example is the Ba₃CoSb₂O₉(PRL 109, 267206 2012; PRL 108, 057205 2012) with equilateral triangular lattice showing 1/3Ms plateau.

R(2) Our study focuses on quantum plateaus, which means that localization of emergent bosonic excitation and no classical counterpart exists. In this regard, the triangular lattice antiferromagnet has little relevance because 1/3 plateau in the triangular lattice antiferromagnet is triggered by “order by disorder” effect; quantum fluctuation stabilizes a classical collinear spin configuration called up-up-down state. In order to clarify this difference and to cite those important experimental observations of plateaus, we revise the introduction and add the third paragraph.

Response to Reviewer #3:

Thank you for the positive comment.

“The experimental results reported in this manuscript are very interesting and should attract attentions from the condensed matter community (in particular, those researchers who work on frustrated magnets).”

Below we reply to his/her comments and questions.

Q(1) Were there any ab initio calculations performed for this material? This would indicate how large J_2 and J_d would be.

R(1) It is difficult to carry out ab initio calculations such as density functional theory calculations because of the occupational disorder of the nitrate ion.

Q(2) Are there any indications from the crystal structure that J_2 and J_d are important? The lengths and angles of the bonds in this material have been reported in Ref. 29, which may give some hints.

R(2) In CdK, superexchange interactions account for dominant magnetic interactions. The J_2 originates from super-superexchange paths and J_d is mediated through Cu-OH-Cd-OH-Cu pathways. A structural consideration may be helpful for a simple Cu-O-Cu path but not for these interactions. Thus, we refer to the case of Kapellasite, whose structure can be seen as Cd^{2+} of CdK is replaced by Zn^{2+} . It is known for Kapellasite that the J_2 is almost zero and J_d is as large as 12 K (B. Fak et al., PRL 109, 037208 2012 and ref. 41). Thus, CdK is supposed to have a small but finite J_d from this analogy. We add the following description to Supplementary 2:

Kapellasite is an isostructural compound of CdK that features relative large J_d of 12 K and negligible J_2 ^{32, 45}.

Q(3) My main concern is that the claim of necessary longer-range interactions (J_2 and J_d) are not fully justified by the results reported in the manuscript and, in my opinion, it is possible to achieve better results by using the numerical technique that the authors already have. Let me explain in more details below.

In Fig. 5, the authors proposed a possible series of magnon crystals for explaining the experimentally observed magnetization plateaus in Fig. 4. It is also suggested that ferromagnetic J_2 and antiferromagnetic J_d interactions would help to stabilize the magnon crystals in Fig. 5.

Did the authors perform PEPS simulations based on the modified kagome model including J , J_2 , J_d , and D terms? I would be surprised if this has not been done, because the authors have calculated the magnetization curve for the Heisenberg model with DM interactions (see Fig. 1). Why cannot the authors calculate the magnetization curve for the modified kagome model including J , J_2 , J_d , and D terms? (By the way, the bond dimension $D=4-7$ in the PEPS simulations performed for obtaining Fig. 1 is pretty small compared to other PEPS simulations in the literature, e.g., those in Ref. 37. What is the reason for that?)

R(3) First let us address on the last question. The main purpose of the present PEPS calculations is to show that the magnetization plateau states, at least the 1/3 plateau, of the nearest-neighbor Kagome Heisenberg model can be viewed as a magnon crystal and that it disappears when we add a large Dzyaloshinskii–Moriya interaction. We think that these

goals have been already achieved by the calculation at the bond dimension $D = 7$, presented in the manuscript, as it is in good agreement with the results for smaller D values (not presented in the manuscript).

As for the further neighbor interactions, of course, if we could include them, as the referee suggested, our scenario about the origin of the successive magnetization plateaus would be reinforced to a certain extent. However, unlike exact-diagonalization or Monte Carlo simulation, the computational cost of the PEPS calculation strongly depends on the lattice structure. Including J_2 and J_4 is far from straightforward in the present PEPS framework and would require a new code, because the present PEPS code heavily relies on the lattice structure where every site is connected only to its nearest neighbors. Thus, we considered, and still consider that the PEPS calculations presented in the manuscript is adequate for the scope of the present paper.

Q(4) In the last sentence of the abstract, the authors claimed that their observation “reveals a novel type of particle physics realized in a highly frustrated magnet”. This is quite confusing. Why is it related to particle physics? I would say that magnons are just ordinary quasi-particles in solid state physics.

In the last paragraph of the main text, it is written that “we are now challenging a magnetization measurement at extremely high fields of 600T, We think that these results will offer not only new physics in frustrated magnetism but also new insights into ...”. In my opinion, these claims are irrelevant for the present results (note that this is not a research proposal) and should be removed from the manuscript.

R(4) We agree to the referee’s opinion and omit these parts.

REVIEWERS' COMMENTS:

Reviewer #1 (Remarks to the Author):

The author has addressed the comments adequately and I recommend publication.

Reviewer #3 (Remarks to the Author):

In this resubmission, the authors have properly addressed the referees' comments/suggestions. The revised manuscript is well written and reports important results. I hence warmly recommend their manuscript for publication in Nature Communications.